# RODIN: INJECTING 2D FOUNDATIONAL FEATURES TO 3D VISION LANGUAGE UNDERSTANDING

## ABSTRACT

We present RODIN (Referential ODIN), a novel model for 3D vision-language understanding that directly operates on posed RGB-D frames. Consuming posed RGB-D from sensors, such as those from an iPhone, simplifies and speeds up inference compared to existing models that train and test using pointclouds sampled from a reconstructed mesh provided by a dataset. We hypothesize that existing approaches consume pointclouds sampled from mesh instead of sensor RGB-D point clouds due to inaccurate camera poses in existing 3D grounding benchmarks, and show that using the "sensor" pointclouds indeed leads to a 5-10% drop in performance on 3D referential grounding, for these methods. Yet sensor noise is unavoidable in real-world settings. RODIN instead addresses this with a scalable, end-to-end architecture for various 3D vision-language tasks. Specifically, RODIN combines powerful pretrained 2D weights trained on internet-scale data, adapts them to a 2D-3D encoder using the recently proposed ODIN, and combines that backbone with a proposed 3D mask-language decoder based on the Mask2Former used in SAM. RODIN achieves state-of-the-art performance on multiple 3D vision-language benchmarks, including referential grounding (SR3D, NR3D, ScanRefer), language prompted object detection (ScanNet200 and Matterport3D), and question-answering (ScanQA and SQA3D). It outperforms previous methods for 3D vision-language tasks, despite consuming only sensor inputs. Because of its combination of effectively leveraging 2D pretrained architectures and finetuning end-to-end on sensor data, RODIN provides a scalable solution for embodied 3D perception.

## 1    INTRODUCTION

The ability to understand and interact with 3D environments through natural language is a cornerstone capability for embodied perception, with applications ranging from robotics to augmented reality. However, a critical challenge in 3D vision-language understanding is the limited availability of large-scale, annotated 3D datasets. While 2D vision models like DINOv2 (Oquab et al., 2024) or CLIP (Radford et al., 2021) benefit from pre-training on millions or billions of diverse internet images, 3D vision-language models (Schult et al., 2023; Lai et al., 2023) are often constrained to small datasets (Achlioptas et al., 2020; Chen et al., 2020) with only thousands of samples on even fewer underlying 3D scenes. This disparity in data scale has led to a significant performance gap between 2D and 3D vision-language models (Majumdar et al., 2024).

Given the power of pre-trained 2D models, a key question emerges: How can we leverage these models, trained on vast datasets, to improve 3D referential grounding? A challenge lies in the misalignment between images, camera poses, and mesh-sampled pointclouds in existing 3D datasets (Kundu et al., 2020a) (e.g. ScanNet (Dai et al., 2017)). This misalignment makes 2D approaches appear less effective on 3D benchmarks, which typically evaluate on point clouds sampled from reconstructed and post-processed meshes. Our experiments show that existing 3D approaches suffer a 5-15% performance drop when trained on "sensor pointclouds" created using raw sensor data, rather than sampling points from reconstructed meshes.

While recent efforts have attempted to port 2D foundational image features to 3D scene understanding (Jatavallabhula et al., 2023; Peng et al., 2023b; Takmaz et al., 2023; Ha & Song, 2022; Tsagkas et al., 2023; Ding et al., 2023; Kerr et al., 2023; Siddiqui et al., 2023; Robert et al., 2022), existing

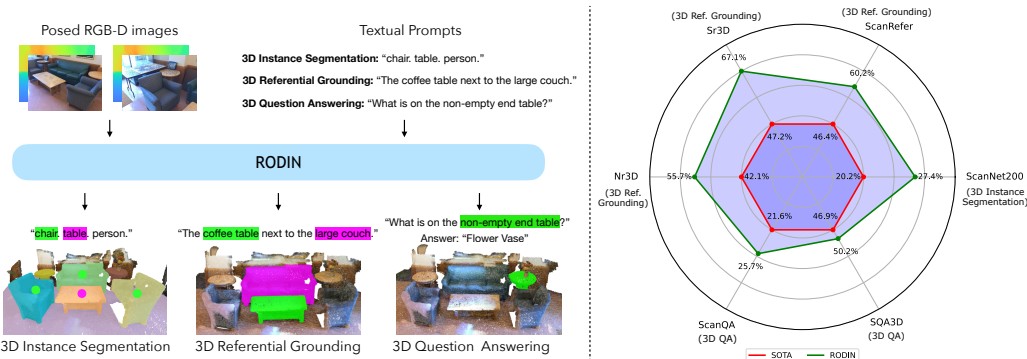

Figure 1: **Left:** RODIN is model for 3D vision-language understanding from posed RGB-D image sequences. **Right:** By carefully designing the architecture to reuse 2D pretrained weights, while also introducing a pathway to learn to be robust to sensor noise, RODIN achieves state-of-the-art performance in 3D referential grounding, language-prompted 3D instance segmentation and 3D question answering benchmarks without using reconstructed meshes.

3D VLMs using 2D pretrained features either don't address 3D language grounding or show poor performance in standard referential grounding benchmarks (Hong et al., 2023b), questioning the usefulness of 2D feature pretraining.

In this paper, we demonstrate that effectively injecting 2D foundational features in 3D vision language understanding is largely a question of architecture: choosing appropriate 3D finetuning and object decoding strategies compatible with 2D feature pretraining.

We propose RODIN (Referential ODIN)[†], a model for 3D referential grounding, question answering and instance segmentation, that effectively incorporates 2D foundational features into 3D vision language understanding. RODIN extends ODIN (Jain et al., 2024) with a novel mask-language decoder for 3D segmentation and a text decoder for question answering. By initializing all encoders and decoders with strong pretrained models, RODIN achieves state-of-the-art performance across various 3D vision-language tasks, using only sensor inputs.

RODIN shows state-of-the-art performance on a broad range of 3D vision-language tasks: on 3D Referential grounding benchmarks of SR3D, NR3D (Achlioptas et al., 2020) and ScanRefer (Chen et al., 2020), 3D segmentation benchmarks of ScanNet200 (Rozenberszki et al., 2022) and Matterport3D (Chang et al., 2017) and 3D Question Answering benchmarks of ScanQA (Azuma et al., 2022) and SQA3D (Ma et al., 2022). We set new state-of-the-art in all referential grounding benchmarks and report significant boost over all prior methods (19.9% on SR3D, 13.6% on NR3D, 13.8% on ScanRefer), as well over a straightforward application of ODIN in referential grounding (+24.1% on Avg). In 3D segmentation benchmarks, we outperform the prior language-prompted models (by 7.2% in ScanNet200). We also outperform all prior SOTA methods in 3D Question Answering (by 4.1% in ScanQA and 3.3% in SQA3D).

In summary our contributions are:

- The first end-to-end model that leverages pretrained 2D features and finetunes them for 3D vision-language reasoning in object detection, referential grounding and question answering.

- Addressing and benchmarking these tasks while eliminating the need for GT boxes and mesh-sampled pointclouds as inputs. The only inputs come from the sensors themselves, making the model applicable for embodied 3D vision.

- State-of-the-art performance on referential grounding, language-prompted object detection and question answering datasets, while also using sensor inputs.

- Through systematic ablations, we demonstrate the superiority of predicting segmentation masks over bounding boxes for 3D language grounding and analyze critical architectural choices for box decoding and mask decoding heads.

---

[†]We pronounce it "road-anne", as in Auguste Rodin, the French sculptor most famous for *The Thinker*.

We believe RODIN opens a path to architectures of 3D vision language models that exploit aspects of "embodiment" such as camera pose and depth and combine this with the robustness of large scale image feature pre-training of foundational 2D VLMs.

## 2 RELATED WORK

**3D Language Understanding Benchmarks**    3D Language Grounding is the task of localizing the objects mentioned in a language utterance using observations of a 3D scene Chen et al. (2020); Achlioptas et al. (2020). This task is primarily studied in the popular benchmarks of SR3D Achlioptas et al. (2020) containing programatically generated sentences, and NR3D Achlioptas et al. (2020) and ScanRefer Chen et al. (2020), containing human-annotated sentences, and 3D scenes from the ScanNet Dai et al. (2017) dataset. The original benchmarks of SR3D and NR3D provide access to ground-truth bounding boxes of all objects in the scenes as input, and the task is to select the correct bounding box, that corresponds to the language sentence. Most methods operate under this assumption, except for BUTD-DETR Jain et al. (2022a), which proposed directly predicting 3D bounding boxes instead of selecting from the available proposals. We follow BUTD-DETR and report results without assuming access to ground-truth boxes. The ScanRefer benchmark is similar to NR3D but does not provide ground-truth boxes as input.

Recently, ScanQA Azuma et al. (2022) and SQA3D Ma et al. (2022) introduced 3D Question Answering Benchmarks. ScanQA focuses on spatial relations. Alongside question-answer pairs, it also includes annotations for the objects referenced in the question. SQA3D Ma et al. (2022) provides pairs of situation descriptions and questions regarding embodied scene understanding, navigation, common sense and multi-hop reasoning, such as *"looking for some food in the fridge"*, *"which direction should i go?"* and the task is to generate the correct answer (*"right"*).

All these benchmarks use point clouds derived from the 3D meshes provided by ScanNet Dai et al. (2017). These meshes were constructed using several steps of post-processing over the raw sensor RGB-D data (which takes minutes-to-hours). These post-processing steps include mesh reconstruction and camera pose estimation, as well as several manual post-processing steps. These processes create fine-grained misalignments between the reconstructed mesh and the sensor RGB-D stream, resulting in drop in performance for methods operating over sensor RGB-D streams instead of the mesh point clouds, as also shown by prior works Robert et al. (2022); Kundu et al. (2020b); Jain et al. (2024). This discourages the uses of sensor RGB-D streams and thus the 2D features pre-trained on internet scale data. In this work, we propose the first 3D language grounding model that operates directly over only sensor RGB-D point clouds. For fair comparison, we benchmark other prior works with sensor point clouds as inputs, and show the benefits of using 2D pre-trained features for 3D language understanding tasks. Using sensor point clouds directly is an emerging idea in the community, further bolstered by the recent introduction of datasets like EmbodiedScan Wang et al. (2023) which also use sensor data directly instead of using meshes.

**3D Visual Language Understanding Models**    3D Visual Grounding Models can be broadly divided into two categories: Two-stage methods and single-stage end-to-end methods. Two stage methods first generate 3D object proposals and then select one proposal out of them. This is the dominant paradigm: InstanceRefer Yuan et al. (2021a), SAT-2D Yang et al. (2021a), ViL3DRel Chen et al. (2022) and recently scaled-up to models of 3DVista Zhu et al. (2023b) and PQ3D Zhu et al. (2024b) which train their model on multiple 3D datasets and tasks. Specifically, 3DVista first pre-trains their model on masked language/object modeling and scene-text matching, and then fine-tunes to downstream several language understanding tasks of interest. PQ3D Zhu et al. (2024b) proposes promptable object queries for 3D scene understanding. While it decodes masks for instance segmentation tasks directly, it follows a 2D stage approach for free-form language grounding and selects a mask from a set of object mask proposals. However, two-stage methods are limited by the failures of the object proposal networks. To overcome this limitation, single-stage methods like 3D-SPS Luo et al. (2022) and BUTD-DETR Jain et al. (2022a) directly regress 3D bounding boxes. They achieve strong results, especially on benchmarks like ScanRefer, which do not provide ground-truth proposals. However, they have only been trained on individual tasks and datasets and haven't been scaled up yet. In this work, we propose a single-stage end-to-end model that is jointly trained on multiple 3D language understanding tasks similar to PQ3D, and achieve state-of-the-art results on several benchmarks. For 3D question answering and captioning, approaches like PQ3D Zhu et al.

(2024b) and 3D-Vista Zhu et al. (2023b) use small text generation heads on top of their language-contextualized features or queries to decode answers. Other approaches like 3D-LLM Hong et al. (2023a) and NaviLLM Zheng et al. (2024) condense the visual scene features into a set of latent vectors and pass it to large pre-trained LLMs like BLIP2-flant5 Li et al. (2023) or Vicuna-7B-v0 Peng et al. (2023a). However, unlike 3D-Vista and PQ3D, they either get significantly poor performance on 3D referential grounding tasks (3D-LLM) or skip evaluating in that setup (NaviLLM). Some very recent efforts from LLAVA-3D Zhu et al. (2024a) make progress towards improving referential grounding performance for the LLM-based methods. In this work, we follow PQ3D and 3DVista's approach and use a small text generation head, mainly for its simplicity.

**Use of 2D Feature for 3D Visual Language Understanding Tasks**   Most 3D Visual Language models directly operate over the provided 3D point clouds without using any 2D pre-trained features. SAT-2D Yang et al. (2021a) is one of the first 3D visual grounding model which used 2D visual features during training for aligning 2D and 3D visual features and show significant boost over its versions that do not use 2D features. Recent methods in 3D Question Answering like 3D-LLM Hong et al. (2023a) and NaviLLM Zheng et al. (2024) use multi-view 2D features and pass them to LLMs for decoding answers. However, as mentioned before, so far they haven't been able to successfully address 3D visual grounding tasks. PQ3D Zhu et al. (2024b) uses a combination of several visual backbones, including a 2D based feature backbone from OpenScene Peng et al. (2023b). Recent work of EFM3D Straub et al. (2024) uses 3D feature volumes obtained from lifting 2D image features but only evaluates on the task of 3D object detection and surface reconstruction. ODIN Jain et al. (2024) proposes an interleaved 2D-3D backbone that utilizes pre-trained 2D weights, but is limited to object detection settings, and as we show in our experiments, does not directly work on 3D language grounding. We extend ODIN to 3D language understanding tasks by proposing architectural changes in its mask decoder head and additional losses to regularize the training.

## 3 METHOD

We show the architecture of RODIN in Figure 2. It is a 2D-3D vision language transformer that accepts a varying number of posed RGB-D images along with a language utterance and fuses information across vision and language streams to predict 3D object segments or generate answers. We featurize the input posed RGB-D images with alternating 2D-3D relative attention backbone proposed in ODIN Jain et al. (2024) and decode 3D object segments with a query based mask decoder that we propose. We next describe the individual modules of this end-to-end differentiable architecture.

**Visual Encoder:** We process the RGB-D input images with ODIN's backbone to obtain a 3D feature cloud for the scene. ODIN is an instance segmentation model for 2D images and RGB-D posed frame sequences. Using the provided depth, ODIN lifts each image in 3D, assigning to each image 2D patch a corresponding 3D point coordinate. It uses a pretrained 2D feature backbone such as ResNet or SWIN, and interleaves 3D attention blocks between 2D residual or 2D attention blocks. To encode 3D relationships in a translation invariant way without a high computational burden, ODIN uses 3D $k$-NN attention layers with relative positional embeddings to fuse information in a geometry-aware way across the input RGB-D views.

**Language Encoder:** We tokenize the text using the language encoder in CLIP Radford et al. (2021). We use an off-the-shelf noun chunker to localize noun phrases in the input utterances.

**Query Refinement Module:** We iteratively update a set of object queries given visual and language features. Our query update mechanism and segment decoding from queries is inspired by Mask2Former Cheng et al. (2022), but substantially differs as their model does not consider language input and only performs closed-vocabulary 2D instance segmentation.

We initialize a set of $M$ learnable object queries, each responsible for decoding an object instance. We concatenate these object queries with the language tokens along the sequence dimension. We alternate between cross-attention between these and the visual feature tokens and self-attention among these concatenated queries and text tokens. Instead of using vanilla cross-attention layer, we follow Mask2Former and use a masked variant where each query only attends to the points falling within the corresponding instance mask predicted by the previous layer. The visual tokens from the backbone are then updated by cross-attending to the updated object and text tokens. Specifically, let

Figure 2: **RODIN Architecture**: 2D 3D vision language transformer that accepts a varying number of posed RGB-D images along with a language utterance and fuses information across vision and language to predict 3D object segments or generate answers. It uses the ODIN backbone that alternates between 2D within image attentions and 3D cross image attentions to produce a 3D feature cloud for the scene in multiple spatial resolutions. The proposed decoder then iteratively updates a set of learnable slot queries as well as the 3D feature tokens though token - language - query attentions to decode object segments and match them to noun phrases in the input referential utterance. A text decoder predicts answers for the input questions through conditioning on the set of updated object queries.

$Q^{(0)} \in \mathbb{R}^{M \times D}$ be the initial object queries, $T \in \mathbb{R}^{L \times D}$ be the text tokens, and $V^{(0)} \in \mathbb{R}^{N \times D}$ be the 3D visual tokens. The query refinement process can be described as follows:

$$X^{(0)} = [Q^{(0)}; T]; \tag{1}$$

$$X^{(i+1)} = \text{Norm}(\text{MaskedCrossAttention}(X^{(i)}, V^{(i)}) + X^{(i)}) \tag{2}$$

$$X^{(i+1)} = \text{Norm}(\text{SelfAttention}(X^{(i+1)}) + X^{(i+1)}) \tag{3}$$

$$V^{(i+1)} = \text{Norm}(\text{CrossAttention}(V^{(i)}, X^{(i+1)}) + V^{(i)}), \tag{4}$$

where $[;]$ denotes concatenation along the sequence dimension, and $i$ is the layer index. The refined queries after each decoder layer $Q^{(i+1)} = X_{1:M}^{(i+1)}$ are then used for mask prediction with the updated visual features and for language grounding.

**Mask Decoder:** The refined object queries decode object segments through a token-wise dot-product with the updated visual features to produce mask logits which are then thresholded to obtain segmentation masks:

$$M_i = \sigma(\text{sigmoid}(Q_i^{(f)} \cdot V^T)), \tag{5}$$

where $M_i$ is the mask for the $i$-th object query, $\sigma$ is a threshold function, and $\cdot$ denotes dot product.

Open-vocabulary mask decoder heads of ODIN and X-Decoder Zou et al. (2023) which also extend Mask2Former's decoder to accept language tokens do not update the visual features during query refinement as we do. We show in our experimental section that while this suffices for 3D instance segmentation, it significantly hurts performance when grounding more complex language in 3D referential grounding (Table-4). Object2Scene Zhu et al. (2023a) shows that updating visual features is unnecessary for decoding 3D bounding boxes, and it is sufficient to only update the queries. We systematically study this in our ablations, and find that updating visual features during query refinement is not necessary for decoding boxes, but is essential for decoding masks through token-wise dot-products (Table-5b).

**Text Decoder:** Beyond decoding segments, the refined object queries are used as input to the decoder of a pre-trained T5 Raffel et al. (2020) to generate answers to questions, following PQ3D Zhu et al. (2024b).

### 3.1 SUPERVISION OBJECTIVE

We match queries to ground-truth instances using Hungarian Matching Carion et al. (2020). We supervise the matched queries's predicted masks with a combination of Binary Cross Entropy (BCE) loss and Dice loss, following Mask2Former. We supervise the inner product between matched queries and visual tokens that belong to the corresponding ground-truth 3D mask with a Binary Cross Entropy loss.

Similar to GLIP Li et al. (2022), MDETR Kamath et al. (2021) and BUTD-DETR Jain et al. (2022a), we match the predicted 3D object segmentations to the relevant noun phrases in the input utterance through a dot-product between a transformation of the object queries and the language tokens, generating a probability distribution $G_i$ over the input text sentence for the $i$th query:

$$G_i = \text{sigmoid}(f_\phi(Q_i^{(f)}) \cdot f_\theta(T^T)) \tag{6}$$

where $f_\phi$ and $f_\theta$ are MLPs, $G_i$ is the grounding probability distribution for the $i$-th object query over the input text tokens. We supervise these grounding distributions with a binary cross-entropy loss. The unmatched queries are supervised to have low-probability over all text tokens.

We observe that our model, trained with the aforementioned objectives, exhibits a failure mode where a small number of distant, unrelated, points are predicted as part of a mask or where multiple instances of the same object category are predicted by a single object query (see Figure 4 in Appendix). As we compare to prior work which evaluates on bounding boxes, this behavior results in oversized bounding boxes. To mitigate this, we find the enclosing 3D bounding box for each mask and supervise using standard box prediction losses (L1 and Generalized Intersection-over-Union Rezatofighi et al. (2019)) against the ground-truth bounding boxes. We add this additional cost both in hungarian matching as well as in the final loss objective. By doing so, we encourage the model to produce more accurate masks which results in more compact bounding boxes, improving performance on downstream tasks.

In summary, our complete loss function reads:

$$\mathcal{L}_{\text{total}} = \lambda_{\text{mask}}\mathcal{L}_{\text{mask}} + \lambda_{\text{text}}\mathcal{L}_{\text{text}} + \lambda_{\text{gen}}\mathcal{L}_{\text{gen}} + \lambda_{\text{box}}\mathcal{L}_{\text{box}} \tag{7}$$

where $\mathcal{L}_{\text{mask}}$ is the mask loss comprised of binary cross entropy and dice losses, $\mathcal{L}_{\text{text}}$ is the loss for matching the object queries to the mentioned objects in the language sentence, $\mathcal{L}_{\text{gen}}$ is the cross-entropy loss over the auto-regressively generated answer (in case of question-answering datasets), and $\mathcal{L}_{\text{box}}$ is the additional bounding box loss described earlier.

**Implementation details:** RODIN consists of 130M parameters, trained end-to-end, with the exception of a frozen 220M parameter text-encoder. We initialize our model with pretrained Mask2Fomer-Swin Liu et al. (2021) trained on 2D data from COCO, with the 3D attention layers of ODIN backbone and additional visual features-to-queries layer in mask decoder from scratch. We train all parameters in data-parallel across 32 A100 80G GPUs with an effective batch size of 64 and otherwise follow the training hyperparameters from ODIN Jain et al. (2024). During training, we process a sequence of $N$ posed RGB-D images. We compute lightweight CLIP embeddings for all images and captions and use this to select 5 relevant frames, with an additional 10 frames coming from Furthest-Point-Sampling (FPS) in CLIP feature space, for a total of 15 frames per 3D scene. At test time, we feed all images in the scene to our model. We use Jina-CLIP Koukounas et al. (2024) as the text-encoder, as it supports arbitrary input-length. We train our model jointly on all datasets, with text generation loss only active in question answering datasets. Our method provides for fast inference, with a 90-frame scene taking ∼1050ms and ∼15GB of VRAM on an A100 GPU.

## 4 EXPERIMENTS

We evaluate our model on 3D referential grounding, 3D object segmentation and 3D question answering. We train a single model across all tasks and benchmarks. Specifically, we train on the 3D

referential grounding datasets of SR3D, NR3D Achlioptas et al. (2020) and ScanRefer Chen et al. (2020); 3D instance segmentation datasets of ScanNet200 Rozenberszki et al. (2022) and Matterport3D Chang et al. (2017); and Question-Answering datasets of ScanQA Azuma et al. (2022) and SQA3D Ma et al. (2022). This is similar in scale to datasets used by prior SOTA methods like PQ3D Zhu et al. (2024b) and 3DVista Zhu et al. (2023b).

## 4.1 EVALUATION ON 3D REFERENTIAL GROUNDING

We use two evaluation setups, following BUTD-DETR Jain et al. (2022a): 1. `Det`, where our model and baselines do not have access to ground-truth 3D boxes of objects in the scene, and 2. `GT`, where our model and baselines use ground-truth 3D object proposals provided in the benchmarks. We evaluate all methods on benchmark-provided point clouds sampled from the post-processed mesh (*Mesh*), and separately we retrain and evaluate a subset of methods on sensor pointclouds (*Sensor*) constructed by unprojecting posed RGB-D images.

Table 1: **Results on 3D language grounding in 3D mesh and sensor point clouds (PC).** We evaluate top-1 accuracy on the official validation set with assuming ground-truth (`GT`) or without assuming ground-truth proposals (`Det`).

| | Method | SR3D | | | | NR3D | | | | ScanRefer | | |
|---|---|---|---|---|---|---|---|---|---|---|---|---|
| | | Acc @25 (Det) | Acc @50 (Det) | Acc @75 (Det) | Acc (GT) | Acc @25 (Det) | Acc @50 (Det) | Acc @75 (Det) | Acc (GT) | Acc @25 (Det) | Acc @50 (Det) | Acc @75 (Det) |
| Mesh PC | ReferIt3DNet Achlioptas et al. (2020) | 27.7 | - | - | 39.8 | 24.0 | - | - | - | 26.4 | 16.9 | - |
| | ScanRefer Chen et al. (2020) | - | - | - | - | - | - | - | - | 35.5 | 22.4 | - |
| | InstanceRefer Yuan et al. (2021b) | 31.5 | - | - | 48.0 | 29.9 | - | - | - | 40.2 | 32.9 | - |
| | LanguageRefer Roh et al. (2022) | 39.5 | - | - | 56.0 | 28.6 | - | - | - | - | - | - |
| | SAT-2D Yang et al. (2021b) | 35.4 | - | - | 57.9 | 31.7 | - | - | - | 44.5 | 30.1 | - |
| | BUTD-DETR Jain et al. (2022b) | 52.1 | - | - | 67.0 | 43.3 | - | - | 54.6 | 52.2 | 39.8 | - |
| | 3D-ViSTA Zhu et al. (2023b) | 56.5 | 51.5 | 42.8 | 76.4 | 47.7 | 42.2 | 35.5 | 65.1 | 51.0 | 46.2 | 36.7 |
| | PQ3D Zhu et al. (2024b) | 62.0 | 55.9 | 46.2 | 79.7 | 52.2 | 45.0 | 37.6 | 66.7 | 56.7 | 51.8 | 43.3 |
| Sensor PC | BUTD-DETR Jain et al. (2022a) | 43.3 | 28.9 | 6.58 | - | 32.2 | 19.4 | 3.64 | - | 42.2 | 27.9 | 6.53 |
| | 3D-ViSTA Zhu et al. (2023b) | 47.2 | 43.2 | 36.1 | 61.4 | 42.1 | 37.4 | 32.0 | 54.2 | 46.4 | 42.5 | 36.3 |
| | RODIN(ours) | **67.1** | **58.7** | **46.4** | **78.9** | **55.7** | **45.9** | **37.2** | **65.8** | **60.2** | **51.8** | **43.2** |

**Evaluation Metrics:** We use the standard top-1 accuracy metric. For the `Det` setup, a predicted bounding box is considered correct if its intersection over union (IoU) with the ground truth box is higher than a predetermined threshhold (we use the standard 0.25, 0.5 and 0.75). Since RODIN predicts masks instead of axis-aligned bounding boxes, we simply convert the masks to bounding boxes via taking the extreme corners of the point cloud falling within the mask. For the `GT` setup, we pool visual features inside the given ground-truth masks, and the object queries predict a segmentation mask over the "pooled" feature tokens, one token per object. The prediction is correct if the model selects the feature token corresponding to the ground-truth object.

**Baselines:** We compare our model against the state-of-the-art two-stage methods of 3D-Vista Zhu et al. (2023b) and concurrent work of PQ3D Zhu et al. (2024b); and the SOTA single-stage method of BUTD-DETR Jain et al. (2022a). All two-stage baselines assume access to ground-truth proposals at test-time in the SR3D and NR3D benchmarks; hence we re-evaluate them with predicted boxes coming from SOTA object detector of Mask3D Schult et al. (2023). We also re-train 3D-ViSTA and BUTD-DETR with sensor point cloud. Despite best efforts, we could not manage to re-train PQ3D with sensor point clouds due to their use of multiple backbones, and multi-stage training strategies.

The 3D referential grounding results are presented in Table-7. We draw the following conclusions:

**Performance of all prior SOTA models drop with sensor point cloud as input and without assuming GT boxes**: Both single-stage methods like BUTD-DETR and two-stage methods like 3D-Vista have a performance drop of 5-15% when using sensor RGB-D point clouds as input instead of mesh point-clouds. The sensor point cloud and mesh point clouds have fine-grained misalignments resulting in this drop. Shifting from ground-truth box proposals to a more realistic setup of using predicted box proposals from a SOTA detector results in a drop of 15-20% accuracy.

**RODIN largely outperforms baselines in both sensor and mesh point cloud setups in the setup where methods do not assume GT boxes** Even when RODIN uses sensor pointclouds (which we showed above result in a 5-15% accuracy drop), it still outperforms the baselines that use *mesh*-point cloud as inputs. RODIN dramatically outperforms alternative single stage models, such as BUTD-

DETR, on the stricter IoU threshhold of 0.75, thanks to predicting masks instead of bounding boxes, as later shown in our ablations (Table-5c). In the GT setup as well, RODIN significantly outperforms 3DVista and closely matches the performance of the very recent work of PQ3D in the setup where PQ3D uses mesh point clouds.

We show qualitative results of RODIN in Figure-3 of Appendix.

## 4.2 Evaluation on 3D Instance Segmentation

Table 2: **Evaluation on 3D Instance Segmentation Benchmarks.** (S) and (M) denotes models trained on sensor and mesh point clouds respectively.

(a) **ScanNet200**

| | Model | mAP | mAP25 |
|---|---|---|---|
| Closed Vocabulary | Mask3D Schult et al. (2023) (S) | 15.5 | 24.3 |
| | Mask3D Schult et al. (2023) (M) | 27.4 | 42.3 |
| | PQ3D (closed) Zhu et al. (2024b) (M) | 27.0 | 46.3 |
| | QueryFormer Lu et al. (2023) (M) | 28.1 | 43.4 |
| | MAFT Lai et al. (2023) (M) | 29.2 | 43.3 |
| | ODIN Jain et al. (2024) (S) | **31.5** | **53.1** |
| Language-Prompted | PQ3D (open) Zhu et al. (2024b) (M) | 20.2 | 32.5 |
| | RODIN (Ours) (S) | **30.2** | **49.6** |

(b) **Matterport3D**

| Input | Model | mAP | mAP25 |
|---|---|---|---|
| Closed Vocabulary | Mask3D Schult et al. (2023) (S) | 2.5 | 10.9 |
| | Mask3D Schult et al. (2023) (M) | 11.3 | 23.9 |
| | ODIN Jain et al. (2024) (S) | **14.5** | **36.8** |
| Language-Prompted | RODIN (Ours) (S) | **13.4** | **29.8** |

We test RODIN on 3D segmentation benchmarks of ScanNet200 Rozenberszki et al. (2022) and Matterport3D Chang et al. (2017) for instance segmentation tasks. These benchmarks have a fixed vocabulary of objects (200 classes in ScanNet200 and 160 classes in Matterport3D). SOTA models like ODIN Jain et al. (2024) and Mask3D Schult et al. (2023) train and evaluate in this fixed vocabulary setup by predicting a distribution over the fixed set of classes and supervising with softmax losses. PQ3D Zhu et al. (2024b) evaluates in a language-prompted setup where they supply object names, one object at a time, and gather predictions for all objects in the vocabulary. They compare with a closed-vocabulary version of their model, and find that their language-prompted version is about 7% worse than their closed vocabulary version due to ambiguities in class names confusing CLIP (eg. "chair" and "armchair"; "table" and "desk" are different categories in ScanNet200). We follow PQ3D and evaluate our model in the language-prompted setup. The input to the model is a concatenation of all object classes of the benchmark as a long sentence (eg: "chair. table. sofa. bed. ...."). While PQ3D cannot predict multiple object classes simultaneously, and hence have to supply one object at a time, our model can simultenously decode masks for all objects mentioned in the sentence. The results are shown in Table-2 on the official validation splits of these benchmarks. We observe that RODIN outperforms PQ3D in the language-prompted evaluation setup on ScanNet200.

## 4.3 Evaluation on 3D and Embodied Question Answering

We test RODIN on ScanQA Azuma et al. (2022) and SQA3D Ma et al. (2022) question answering benchmarks. ScanQA Azuma et al. (2022) focuses on spatial relations. Alongside question-answer pairs, the dataset includes annotations for the objects referenced in the question, and we supervise our model to predict them in addition to generating the answer. SQA3D Ma et al. (2022) provides pairs of situation descriptions and questions regarding embodied scene understanding, navigation, common sense and multi-hop reasoning, such as, *"looking for some food in the fridge"*, *"which direction should i go?"* and the task is to generate the correct answer (*"right"*).

**Evaluation Metrics:** We use the established Exact Match (EM@1) metric, which measures if the generated answer matches either of the two provided answer candidates for ScanQA and the single ground-truth answer in SQA3D.

**Baselines:** We compare against the LLM based methods of 3D-LLM Hong et al. (2023a) and Nav-iLLM Zheng et al. (2024) which use BLIP2-flanT5 Li et al. (2023) and Vicuna-7B Peng et al. (2023a) as their answer generation heads. We also compare with 3D-Vista Zhu et al. (2023b) and PQ3D Zhu et al. (2024b) which use small decoder heads like T5-small Raffel et al. (2020) similar to our approach. We show results in Table-3 on the validation sets of these benchmarks. RODIN outperforms all prior baselines on both benchmarks. We found that using sensor point clouds vs mesh point clouds does not result in a significant difference in performance in these benchmarks,

likely because the models are evaluated on text generation instead of localization of objects as in 3D referential grounding and segmentation benchmarks.

Table 3: **Results on Visual Question Answering in 3D Point Clouds** on official validation sets. We evaluate top-1 exact match accuracy (EM@1).

| | Method | ScanQA | SQA3D |
|---|---|---|---|
| Mesh PC | 3D-LLM (BLIP2-flant5) Hong et al. (2023a) | 20.5 | – |
| | PQ3D Zhu et al. (2024b) | 21.0 | 47.0 |
| | 3D-VisTA Zhu et al. (2023b) | 22.1 | **47.5** |
| | NaviLLM Zheng et al. (2024) | **23.9** | – |
| Sensor PC | 3D-VisTA Zhu et al. (2023b) | 21.6 | 46.9 |
| | RODIN (Ours) | **25.7** | **50.2** |

### 4.4 ABLATIONS

Table 4: **Ablations** Acc@25 in `Det` Setup

| Model | Avg Accuracy | SR3D | NR3D | ScanRefer |
|---|---|---|---|---|
| RODIN | **61.0** | **67.1** | **55.7** | **60.2** |
| w/o mask decoder w/ box decoder | 39.3 | 38.9 | 33.2 | 45.7 |
| w/o feature attn | 36.9 | 38.0 | 30.0 | 42.8 |
| w/o pretrained 2D weights | 53.4 | 54.3 | 49.1 | 56.9 |
| w/o mask bounding box loss | 56.8 | 64.3 | 49.5 | 56.7 |

Table 5: **Analysis of Box Head vs Mask Head** on ScanRefer Dataset with Acc@25 if not otherwise stated.

(a) **Parametric vs Non-parametric Query**

| Query Type | Box Head | Mask Head |
|---|---|---|
| Param | 23.9 | **54.4** |
| Non-param | **34.5** | 43.9 |

(b) **Updating Visual Features with Language + Object Queries**

| Feat Attn | Box Head | Mask Head |
|---|---|---|
| ✓ | 33.9 | **54.4** |
| ✗ | **34.5** | 41.5 |

(c) **Results at Various IoU Thresholds**

| | Acc@25 | Acc@75 |
|---|---|---|
| Box Head | 34.5 | 1.1 |
| Mask Head | **54.4** | **33.2** |

We ablate a series of design choices of our model on referential grounding datasets of SR3D, NR3D, and ScanRefer on Table-4 and on ScanRefer dataset in Table-5. We have the following conclusions:

**1. Decoding boxes is inferior to decoding segmentations.** Shifting from decoding segmentation masks to decoding bounding boxes hurts performance (row 2 of Table 4), especially in tight IoU thresholds IoU@0.75, as shown in Table-5c.

**2. Visual tokens updating through attending to language and queries during mask decoding is essential** for good performance in 3D referential grounding, as shown in row 3 of Table-4. This is potentially because the mask decoding head relies on dot-product of queries and features to predict masks; and thus having both object queries and visual features to be very well distinguished for different instances of the same object is crucial. This design choice is unique to mask decoding heads, as we show in Table-5b. Box-decoding models work similarly well irrespective of updating the visual tokens with language and object tokens. This variant is very close to ODIN's open vocabulary head, which also lacks such attentions, and as we show it does not work well for referential language grounding.

**3. 2D feature pretraining dramatically improves performance** as shown in row 4 of Table-4.

**4. The predicted mask bounding box loss helps** as shown in row 5 of Table-4.

**5. Non-parametric queries are crucial for decoding boxes prediction, while parametric queries work well for decoding segments.** There are two popular choices for object queries : *Parametric*

*Queries* which are scene-independent learnable vectors, initialized from scratch, and are updated via attention. *Non-Parametric Queries*, which are scene-dependent, and are typically initialized by doing Furthest Point Sampling on the input point clouds and encoding the corresponding xyz locations as query positional embeddings and corresponding features as query feature embeddings. Box-decoding heads need to regress raw XYZ coordinates in 3D space; the search space is large and sparse—as most of it is empty—and parametric queries have difficulty handling such free space, as already mentioned in 3DETR Misra et al. (2021). Mask decoding uses dot-product between queries and visual tokens coming from 3D backbone, and thus do not need to reason about 3D free space.

**Discussion**: Certain datasets like Arkit3DScenes Baruch et al. (2021) and Aria Datasets Straub et al. (2024) only have supervision available for 3D boxes instead of masks, making box decoding more favourable. However, recent methods like Box2Mask Chibane et al. (2022) show that segmentation predictions can be adequately supervised with bounding box labels as well. While updating visual features via attention to queries and language and an additional box loss help, we still see some outlier points segmented in 3D as well as models predicting multiple instances of the same object as the predicted answer (see Figure-4 of Appendix). When converting masks to bounding boxes as a post-processing step, these errors results in oversized and wrong bounding boxes. Hence, further research is needed to fix these issues of mask decoding heads. Models like ODIN and RODIN attempt to unify 2D and 3D perception tasks, and predicting masks is a common interface across the two– simply dot-product between queries and visual features (either 2D or 3D). Box decoding requires separate prediction heads with either 4D or 6D dimensional outputs for 2D and 3D, respectively. This makes mask-decoding heads preferable for unifying 2D-3D perception tasks.

## 5 CONCLUSION

We presented RODIN, a model for 3D vision-language understanding that operates directly on posed RGB-D images to localize referenced objects and answer questions. RODIN is the first end-to-end model that leverages pretrained 2D features, finetunes them for several 3D vision language tasks and achieve state-of-the-art performance on multiple 3D vision-language benchmarks, including SR3D, NR3D, ScanRefer, and ScanQA and SQA3D, while using only sensor point cloud inputs. We conducted extensive ablations that justify our design choices in decoding segmentations masks, updating visual tokens during object query refinement, the use of pretrained 2D features, and the addition of a mask bounding box loss. We believe RODIN is a simple and scalable 3D vision-language model that fills in a gap in existing 3D vision literature, serving as a general model that exploits 2D feature pretraining while still taking advantage of 3D, with direct RGB-D input. Our future work will explore its extensions to further scaling up these models by exploring joint training with 2D and 3D vision-language understanding tasks and their applications to robot 3D perception, object tracking and embodied scene understanding.

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

# A    APPENDIX

## A.1    EFFECT OF FINE-TUNING 2D BACKBONES IN RODIN

We study the effect of fine-tuning the 2D backbones on in-domain and out-of-domain performance. We train two versions of RODIN, one with fine-tuning and the other without fine-tuning. For training, we use SR3D and NR3D, and evaluate on the validation sets of SR3D, NR3D (in-domain) and ScanRefer (out-of-domain). The results of the experiments are shown in Table-6. We find that both models work similarly well, both in-domain and out-of-domain. In the main paper, we fine-tune all parameters of the model. We note that the out-of-domain experiment is not conclusive as the visual scenes come from ScanNet for both in-domain and out-of-domain datasets.

Table 6: **Effect of Fine-tuning 2D backbones of RODIN** for Acc@25 in `Det` Setup. SR3D and NR3D are in-domain and ScanRefer is out-of-domain

| Model | SR3D | NR3D | ScanRefer |
|-------|------|------|-----------|
| RODIN w/ finetune | 65.6 | 52.7 | 54.4 |
| RODIN w/o finetune | 66.7 | 52.0 | 54.5 |

## A.2    PERFORMANCE WITH DIFFERENT BACKBONES

In this section, we demonstrate that RODIN's method can be applied to multiple backbones and is not dependent on ODIN. We demonstrate that the performance can scale with the strength of the backbone. Specifically, we integrate a DINOv2 Oquab et al. (2024) backbone consisting of 1.1B parameters, scaling over 5x compared to the Swin backbone we use in all other experiments. To achieve high-performance during training, we freeze the backbone, although we note that it is possible that additional performance could be obtained with efficient fine-tuning techniques such as LoRA Hu et al. (2021). We find that adding this backbone boosts performance on all 3 language grounding datasets, with substantial margins of 4.2%, 1.9%, and 3.4% @ 0.25 on SR3D, NR3D, and ScanRefer respectively. These results further demonstrate the impressive results of RODIN, surpassing all prior methods on these 3 language grounding datasets by an average margin of 7.1%, even when comparing to methods evaluate on with a post-processed mesh (PQ3D).

Table 7: Ablation of visual backbones on 3D language grounding. We evaluate top-1 accuracy on the official validation set without assuming ground-truth proposals (`Det`).

| Method | SR3D | | | NR3D | | | ScanRefer | | |
|--------|------|------|------|------|------|------|-----------|------|------|
| | Acc @25 (Det) | Acc @50 (Det) | Acc @75 (Det) | Acc @25 (Det) | Acc @50 (Det) | Acc @75 (Det) | Acc @25 (Det) | Acc @50 (Det) | Acc @75 (Det) |
| RODIN (Swin) | 67.1 | 58.7 | 46.4 | 55.7 | 45.9 | 37.2 | 60.2 | 51.8 | 43.2 |
| RODIN (DINOv2) | **71.3** | **62.9** | **48.9** | **57.6** | **47.5** | **38.2** | **63.6** | **55.2** | **44.6** |

## A.3    ADDITIONAL METRICS ON SCANQA DATASET

We report additional standard metrics used by ScanQA benchmark in Table-8.

## A.4    VISUALIZATIONS OF RODIN ON 3D REFERENTIAL GROUNDING DATASETS

We show the visualization of RODIN in Figure-3.

## A.5    VISUALIZATION OF COMMON FAILURE MODES OF RODIN

We identify three systematic failure modes in our model, illustrated in Figure-4.

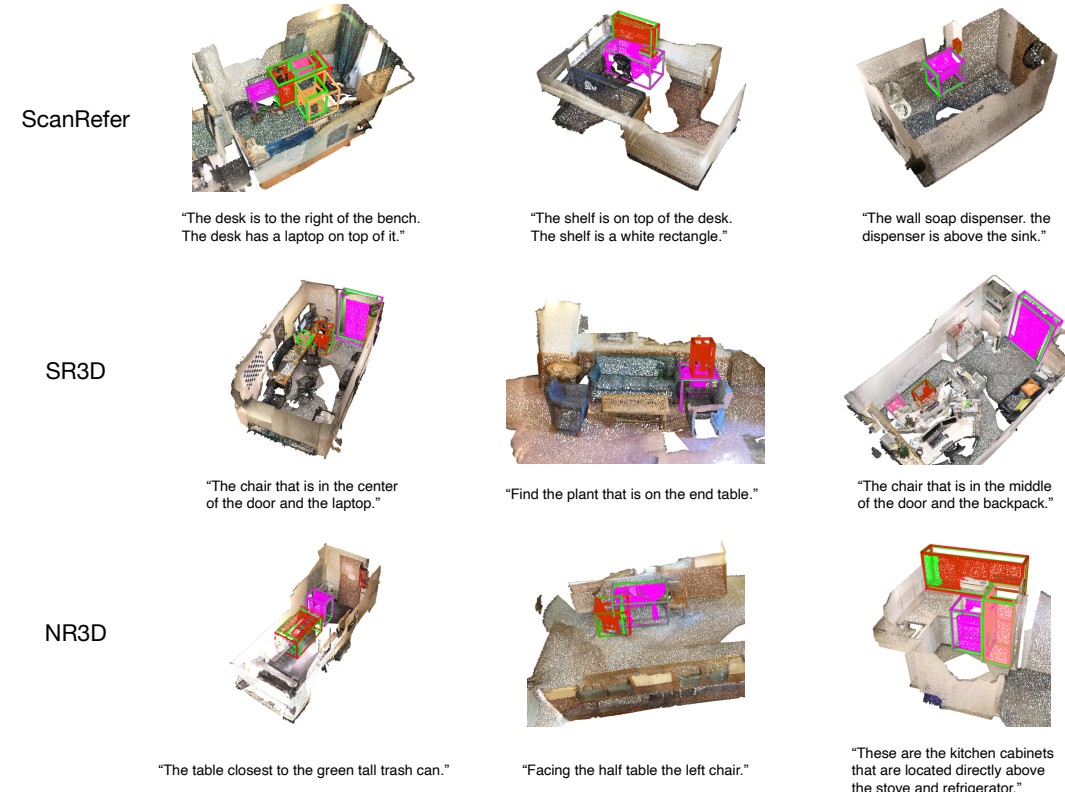

Figure 3: Visualizations of RODIN on 3D Referential Grounding Datasets of ScanRefer, SR3D, and NR3D

- **Inclusion of distant outlier points in the predicted masks**: In the first image of Figure-4, while RODIN accurately predicts the object, it also mistakenly includes some distant points in the mask. This leads to a larger bounding box during the mask-to-bounding box conversion in post-processing, negatively affecting accuracy metrics. Although our proposed box loss mitigates this issue, it doesn't fully resolve it.

- **Multiple instances of the same object being segmented together**: As shown in the middle image of Figure-4, RODIN predicts both beds as a single output. Incorporating attention to language and queries helps reduce such errors, though they still persist. Our box loss also aids in addressing this issue.

- **Failures in language understanding** as seen in the third image of Figure-4.

The first two failure modes are specific to mask-decoding architectures, and similar issues have been noted by Mask3D Schult et al. (2023) in their 3D instance segmentation tasks. Box-decoding architectures, on the other hand, generally avoid these problems. Nevertheless, we find that mask-decoding architectures offer significant advantages in other aspects, such as more accurate and fine-grained segmentation, making them valuable despite these challenges.

## A.6 DETAILED ARCHITECTURE DIAGRAM OF RODIN

We show a detailed diagram of RODIN, with additional on visual backbone in Figure-5.

## A.7 PERFORMANCE ANALYSIS WITH POSE AND DEPTH NOISE

To analyze the performance of RODIN under sensor noise we conduct two experiments to model error in both pose and depth. For the pose error experiment, we add gaussian noise to the translation and rotation components of every camera pose in a scene. Similarly, for the depth error experiment,

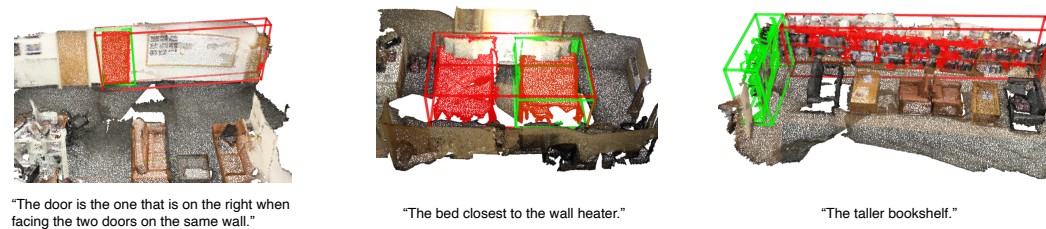

"The door is the one that is on the right when facing the two doors on the same wall."

"The bed closest to the wall heater."

"The taller bookshelf."

Figure 4: Systematic failure modes of RODIN

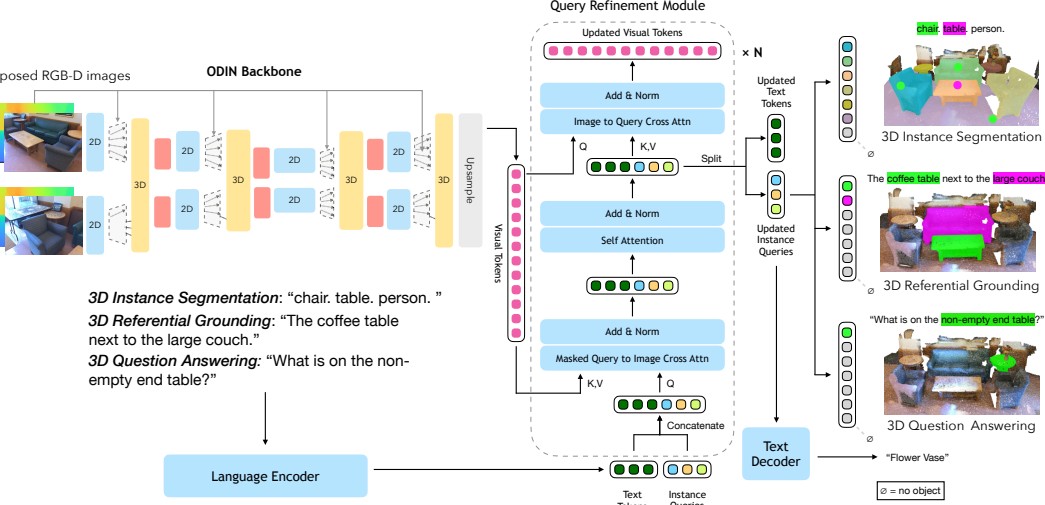

Figure 5: **Detailed RODIN Architecture**: 2D 3D vision language transformer that accepts a varying number of posed RGB-D images along with a language utterance and fuses information across vision and language to predict 3D object segments or generate answers. It uses the ODIN backbone that alternates between 2D within image attentions and 3D cross image attentions to produce a 3D feature cloud for the scene in multiple spatial resolutions. The proposed decoder then iteratively updates a set of learnable slot queries as well as the 3D feature tokens though token - language - query attentions to decode object segments and match them to noun phrases in the input referential utterance. A text decoder predicts answers for the input questions through conditioning on the set of updated object queries.

we add gaussian noise uniformly to the depth map. When each depth map is unprojected, the resulting point cloud becomes misaligned and performance decreases. We use relative pose error as defined in Sturm et al. (2012).

We compare the robustness of RODIN to prior state-of-the-art single-stage method of BUTD-DETR Jain et al. (2022a). We chose a single-stage method as our baseline, since multi-stage methods like PQ3D Zhu et al. (2024b) and 3D-Vista Zhu et al. (2023b) rely on several external models, and use pre-processed intermediate outputs from them for their inference. This makes it harder to fairly run comparisons directly on the point cloud input. As shown in Figure-6, RODIN is highly robust to both types of noise. At a mean error of 0.2, RODIN impressively maintains a Top1@0.25 mIoU accuracy of 66.7%.

In the pose error case, the model must understand the misaligned point cloud and cannot simply ignore the spurious points. However, RODIN still shows impressive robustness with substantially less degradation compared to BUDT-DETR.

We believe a great portion of robustness comes from reliance on 2D pre-trained features and 2D layers in the network. Despite the noise in the depth and pose estimation, they still operate over the clean RGB images. Additionally, our 3D layers use local and relative attentions, which additionally contribute to the robustness.

Table 8: **Extra Metrics on ScanQA validation set**

|  | Method | EM | BLEU-1 | ROUGE | METEOR | CIDEr |
|---|---|---|---|---|---|---|
| Mesh PC | 3D-LLM Hong et al. (2023a) | 20.5 | **39.3** | 35.7 | 14.5 | 69.4 |
|  | PQ3D Zhu et al. (2024b) | 21.0 | - | - | - | - |
|  | 3D-VisTA Zhu et al. (2023b) | 22.5 | 32.0 | 35.5 | 13.8 | 69.1 |
|  | NaviLLM Zheng et al. (2024) | **23.0** | - | **38.4** | **15.4** | **75.9** |
| Sensor PC | 3D-VisTA Zhu et al. (2023b) | 21.6 | 30.1 | 34.1 | 13.2 | 65.3 |
|  | RODIN (Ours) | **25.7** | **36.1** | **40.0** | **15.2** | **78.5** |

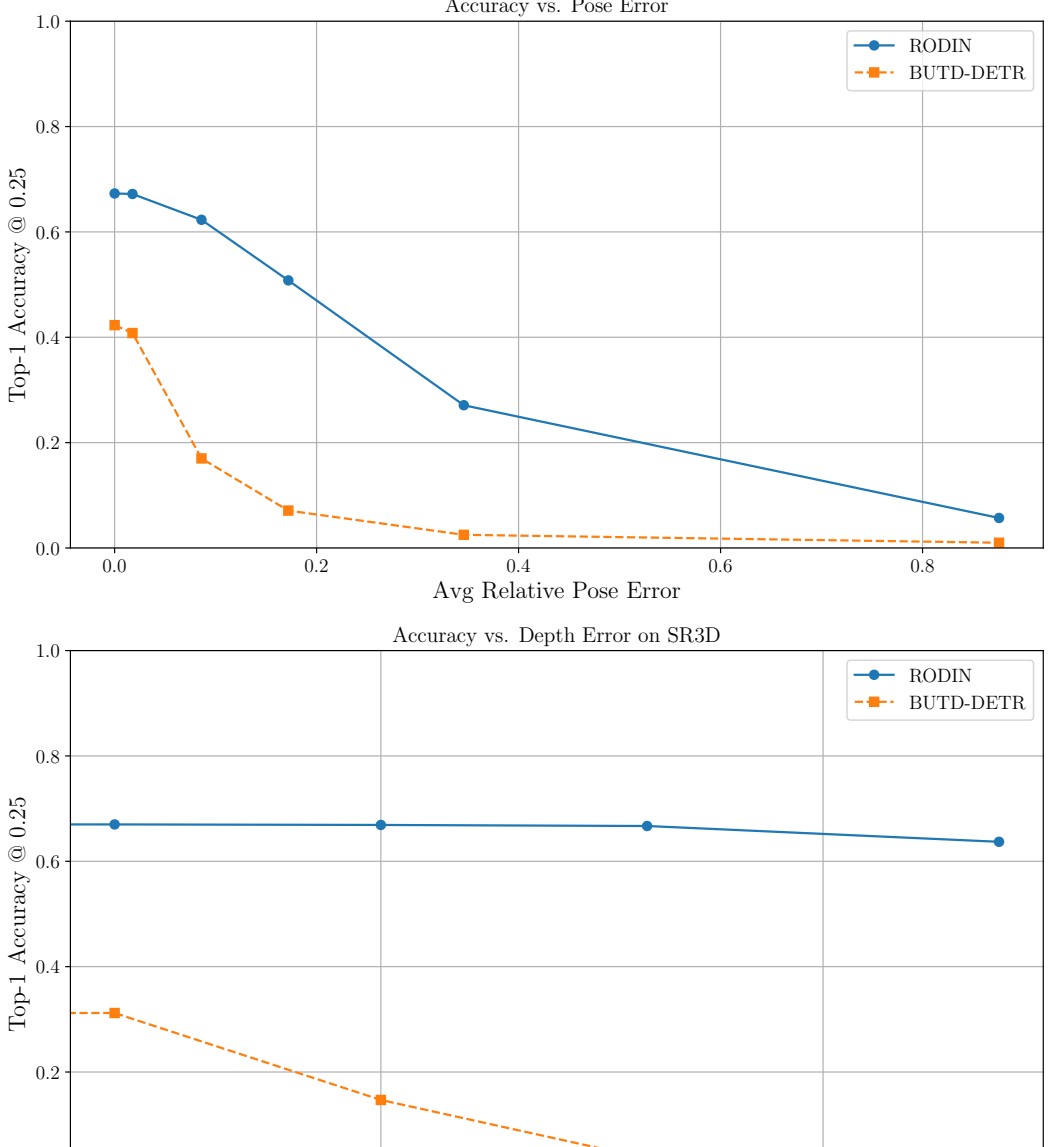

Figure 6: We analyze the performance of RODIN and BUDT-DETR as the pose and depth error increases. We add gaussian noise to the pose and raw depth which affects the unprojected point cloud that both models observe.