# OpenReview forum: "RODIN: Injecting 2D Foundational Features to 3D Vision Language Understanding"
_ICLR.cc/2025/Conference — Submitted to ICLR 2025_

### Official Review · Reviewer_mEfc · 2024-11-02

**Soundness:** 2
**Presentation:** 1
**Contribution:** 2
**Rating:** 5
**Confidence:** 5

**Summary:**

Building on prior work ODIN, RODIN introduces a language branch for 3D vision-language tasks, directly using posed RGB-D frames for 3D grounding, object detection, and 3D QA, thus bypassing mesh-based point clouds. It employs masked cross-attention modules, inspired by Mask2Former, to facilitate interaction between images and language. RODIN’s performance is validated across benchmarks such as SR3D, NR3D, ScanRefer, SQA, and ScanQA.

**Strengths:**

1. The work shows promising performance across Referit3D and ScanRefer.
2. The first end-to-end method that leverages 2D features.

**Weaknesses:**

1. The model primarily appears as a combination of ODIN with an added language branch and cross-modal interaction module, trained on large-scale 3D vision-language (VL) task heads. Compared to ODIN, which also uses RoBERTa as a language encoder, the technical contribution seems focused on engineering, incorporating Mask2Former’s masked cross-attention modules and an additional MLP language head, similar to PQ3D (lines 167-168).

2. The stated contribution, "The first end-to-end model leveraging pretrained 2D features and finetuning for 3D vision-language reasoning in object detection, referential grounding, and question answering," seems weak, as the visual backbone is ODIN-based (lines 195-201) and is not introduced in this paper.

3. The open-vocabulary claim lacks experiments to demonstrate zero-shot capability.

4. Despite incorporating more training data, the model underperforms ODIN on ScanNet-200 and Matterport3D. Moreover, ODIN already uses open-vocabulary detection prompts like GLIP.

5. The paper lacks citations to relevant 2D methods, such as MDETR [3] and GLIP [2].

6. Figure 2 lacks details on the visual encoder design, though it references ODIN, which uses a cross-view encoder.

[1] Jain, Ayush, et al. "ODIN: A Single Model for 2D and 3D Segmentation." CVPR 2024
[2] Li, Liunian Harold, et al. "Grounded language-image pre-training." CVPR 2022
[3] Kamath, Aishwarya, et al. "MDETR: Modulated Detection for End-to-End Multi-Modal Understanding." ICCV 2021

**Questions:**

See the weakness.

1. Missing evaluation on additional ScanQA metrics such as CIDEr and BLEU, which would provide a more comprehensive assessment of performance.

---

> ### Author Response · Authors · 2024-11-21
>
> Thank you for your constructive feedback and suggestions for experiments. We address your concerns point-by-point below.
>
> **Q4.1**
> > The model primarily appears as a combination of ODIN with an added language branch and cross-modal interaction module, trained on large-scale 3D vision-language (VL) task heads. Compared to ODIN, which also uses RoBERTa as a language encoder, the technical contribution seems focused on engineering, incorporating Mask2Former’s masked cross-attention modules and an additional MLP language head, similar to PQ3D (lines 167-168).
>
> We want to clarify that masked cross-attention from object queries to visual features is not our main change – that exists in Mask2Former and is copied forward in RODIN. We add an additional attention from visual features to object queries – this attention layer does not exist in any prior architectures of Mask2Former / ODIN / BUTD-DETR. In Table-4 and Table-5, we clearly ablate and show that this change is critical to performance for referential grounding by improving performance by 24.2% (L477-481). Besides, we add an additional box loss which improves performance by 4.2%. We want to note that  many impactful works are essentially engineering innovations -- MAE is essentially BERT for the visual domain, MaskFormer is DETR with mask decoding heads, ODIN is Mask2Former with additional 3D attention layers, etc. RODIN relies on generic architectures like ODIN and Mask2Former, and extends it to 3D vision language tasks. It outperforms prior SOTA (with often complex architectures and more engineering) **by more than 10%** across several benchmarks, which highlights the importance of our technical contributions / engineering.
>
> We also want to draw your attention to other contributions of this work besides the technical contributions. As other reviewers also note, we benchmark RODIN and prior SOTA baselines on more realistic settings for robotics/embodied AI and demonstrate even larger performance gains in such real-world settings. RODIN is also the first end-to-end trainable 3D referential grounding method with SOTA performance, bypassing prior methods with multi-stage training.
>
> \
> **Q4.2**
> > The stated contribution, "The first end-to-end model leveraging pretrained 2D features and finetuning for 3D vision-language reasoning in object detection, referential grounding, and question answering," seems weak, as the visual backbone is ODIN-based (lines 195-201) and is not introduced in this paper.
>
> We think that RODIN is indeed the first end-to-end model that achieves SOTA performance for 3D vision-language tasks. As we discuss in L146-170, prior SOTA like 3D-Vista and PQ3D are two-stage methods; single-stage methods like BUTD-DETR and SPS do not work well at high IoU thresholds such as $0.75$ IoU (Table-1) and have never been scaled up to be jointly trained on multiple datasets and tasks. RODIN achieves this performance by using 2D features for which we indeed use ODIN backbone, but design a new mask decoder head.
>
> As previously stated, a direct application of ODIN for referential grounding is worse by 24.2% than RODIN (L477-481). Prior to RODIN, there is no 3D referential grounding model which is both single-stage, utilizes 2D features and obtains state-of-the-art performance – all while operating in realistic setups of sensor point cloud input and no assumptions of GT object proposals.
>
> Last, we have extended RODIN to use DINOv2 ViT backbones for image featurization, without 3D fusion inside the ViT 2D backbone (like the ODIN strategy), but rather, post 3D fusion after the 2D featurization across views. Using ViT 2D backbones improves performance over the initially reported RODIN results on all benchmarks by $4.2\%$, $1.9\%$, and $3.4\%$ @ 0.25 IoU on SR3D, NR3D, and ScanRefer, respectively. Hence, the key message of RODIN is that using 2D features significantly boosts performance for 3D visual language tasks, and it is not restricted to the ODIN backbone.
>
> \
> **Q4.3**
> > The open-vocabulary claim lacks experiments to demonstrate zero-shot capability.
>
> Our submission originally followed PQ3D’s definition of open-vocabulary. Their method supplies a *fixed* set of names as language input to the model and labels this as an open-vocabulary setup. We agree that this is confusing and a non-standard definition, and we delete it. We changed all occurrences of “open-vocabulary” to “language-prompted” in our revised submission. Thank you for pointing this out.

---

> ### Author Response · Authors · 2024-11-21
>
> **Q4.4**
> > Despite incorporating more training data, the model underperforms ODIN on ScanNet-200 and Matterport3D. Moreover, ODIN already uses open-vocabulary detection prompts like GLIP.
>
> ODIN only reports results in the closed-vocabulary setting for ScanNet200 and Matterport3D and thus these results are not directly comparable to ours.
>
> As we discuss in L400-410, the closest to our setup is PQ3D, which shows a 7% drop in performance in language-conditioned input compared to an MLP-based closed-vocabulary setup. In Appendix Section B, they analyze and attribute this drop to ambiguities in class labels of ScanNet200 (like: “cabinet” – “kitchen cabinet”, “chair” – “armchair”, “desk” – “table”) and thus their similarities in the language space of CLIP. We believe this affects our results as well.
>
> Since submission, we have tuned some additional hyperparameters, resulting in RODIN’s performance of 30.2 mAP on ScanNet200 (ODIN’s mAP is 31.5), and 13.4 mAP on Matterport3D (ODIN’s mAP is 14.5), without losing out on performance on referential grounding. RODIN’s performance is in a similar ballpark as ODIN, despite being more general than ODIN, and outperforms PQ3D in a comparable evaluation setup by 10.0% on ScanNet200, while using less 3D data than their method.
>
> \
> **Q4.5**
> > The paper lacks citations to relevant 2D methods, such as MDETR [3] and GLIP [2]. Figure 2 lacks details on the visual encoder design, though it references ODIN, which uses a cross-view encoder.
>
> Thank you for pointing these out. We have added these citations in the revised draft and added an additional diagram in Figure-5 of supplementary with details of visual encoder for completeness.
>
> \
> **Q4.6**
> > Missing evaluation on additional ScanQA metrics such as CIDEr and BLEU, which would provide a more comprehensive assessment of performance.
>
> Thank you for the suggestion. Since submission, we have added additional metrics in Supplementary, Table 8. RODIN outperforms all baselines operating over sensor point clouds by significant margin on all the additional metrics (4.1% EM, 6.0% BLEU-1, 5.9% ROGUE, 2% METEOR, 13.2% CIDEr).

---

> ### Author Response · Authors · 2024-11-26
>
> Thank you for carefully reading our rebuttal and for raising your score. We address few additional comments below:
>
> > “Works like MAE/MaskFormer are noteworthy because they validate the generality of masked image modeling and a universal segmentation head across various classification and segmentation backbones, thoroughly examining the trade-offs in module design. In the same way, the significance of mask-language decoder design (L81-84) should be demonstrated by its integration with existing 3D backbones rather than mere modifications to ODIN.”
>
> Thank you for your suggestion. As the title of our paper suggests, our focus is how to best leverage foundational 2D features for 3D vision-language understanding, which existing 3D backbones, (e.g., PointNet) that operate over colored point clouds do not do. Thus, the focus of our paper is achieving this goal.
>
> In our previous comment, we directly studied the significance of the mask-language decoder design by training our model with an entirely new 2D-3D backbone, composed of frozen 2D ViT backbone (DINOv2) and 3D KNN fusion layers only after the 2D image featurization. This additional experiment demonstrates that the proposed decoder is independent of the backbone, and in particular, the ODIN backbone that we obtained our initial results with. We also note that the ODIN mask decoder—which we demonstrate does not work on this task—is an exact replica of Mask2Former (except they replace 2D positional embeddings with 3D positional embeddings).
>
> Due to the time constraints of the discussion period, we won’t be able to integrate additional backbones with our decoder, but we believe that the RODIN decoder can work with any backbone (e.g., Point Transformers or Minkowski).
>
> > “Besides, my primary concern about the limited research insights persists. Although the paper claims to pioneer the first end-to-end joint 2D-3D models, it closely follows ODIN’s framework with only minor changes, such as the inclusion of cross-attention object queries. This approach appears to be more of a direct extension of ODIN. Despite the rebuttal experiments for Q4.2 suggesting that the joint 2D-3D model of ODIN might not be necessary, it remains unclear whether this gap stems from modifications specific to RODIN's modules or if similar changes to ODIN would exhibit the same behavior.”
>
> We kindly ask you to examine the claim that “RODIN is a minor change over ODIN”. 3D referential grounding is a very active field of research with dozens of papers published on these benchmarks every year. RODIN outperforms all prior works with significant margins (by over 10%) – had this been a minor or an obvious change over ODIN, this baseline would already exist by now. Additionally, if a “minor” change offers significant benefits over more sophisticated methods – that too in more realistic setups than prior works, we argue this is extremely relevant to the community and worthy of dissemination.
>
> > “it remains unclear whether this gap stems from modifications specific to RODIN's modules or if similar changes to ODIN would exhibit the same behavior”
>
> The point of this experiment is to show that multiple reasonable ways of encoding 3D points with 2D foundational features work for 3D referential grounding with our language- mask decoding head, and thus the use of the ODIN backbone is an implementation detail and not required for our results. The gains in the rebuttal experiments come from using stronger ViT backbones instead of Swin, and in our experiments extending ODIN with ViT indeed results in similar performance. We do not claim that this new strategy of 2D-3D featurization is better than ODIN—instead, it is an attempt to show that 2D-3D visual backbone is not our contribution, but rather that using 2D features with our proposed mask decoder result in SOTA performance for 3D visual language grounding.
>
> > “The first and second contributions concerning sensor input and the leveraging of pretrained 2D features predominantly derive from ODIN (L196-L200), which also integrates sensor input and utilizes 2D features. The principal changes involve adaptations to the task head.”
>
> ODIN made that contribution for instance segmentation but not for 3D referential grounding. Its performance on referential expression is 24.2% worse than RODIN (Table-5, L477-481). We additionally benchmark prior SOTA methods by using sensor point clouds on referential grounding tasks (5-10%) and question answering tasks (negligible difference). RODIN is the first successful instantiation of a 3D language grounding model that operates over sensor point clouds and does not use unrealistic assumptions of GT bounding box proposals. We believe that benchmarking in these realistic settings and showing a successful instantiation of a sensor based model like RODIN will make it easier for future works to make progress in settings more aligned to goals of embodied vision.

---

> ### Author Response · Authors · 2024-11-26
>
> > “The fourth contribution more closely resembles an ablation study rather than a standalone contribution to me.”
>
> The fourth contribution is : “Through systematic ablations, we demonstrate the superiority of predicting segmentation masks over bounding boxes for 3D language grounding and analyze critical architectural choices for box decoding and mask decoding heads.”. This is not merely an ablation of RODIN but experimentally elicits a key insight of our work: Mask decoding heads perform better than box decoding heads for 3D referential grounding. This is relevant for the 3D Visual Language community since all single-stage methods, like BUTD-DETR and 3D SPS decode boxes, and while 3D instance segmentation has been studied for a long time, no prior work decode masks for 3D visual grounding. We particularly find that decoding masks significantly improves performance on higher IoU thresholds of 0.75, where box decoding heads obtain near 0% performance (Table-5). Additionally, we show the additional attention from visual features to object queries is critical for mask decoding heads but not for box decoding heads (while prior works of Object2Scene find that updating visual features is unnecessary – however they never tried decoding masks). In summary, these “ablation studies” question some of the well-established methodologies and design-choices of prior works and discuss key modeling choices, which we believe will help future work.
>
>
> Overall, the current trend in the 3D language grounding community is to:
>
> a) Encode 3D point cloud directly without using 2D features
> b) Operate over reconstructed and post-processed mesh point clouds
> c) Assume access to GT proposals and / or operate in two-stages i.e. first detect boxes and then select the correct proposal
> or d) Not assume GT boxes and operate in single-stage but decode boxes and obtain low performance on strict thresholds.
>
> RODIN challenges these standard practices and obtains strong performance over all prior methods. We kindly ask you to reexamine your rating / decision in context of these broader contributions of our work. We are very happy to engage in discussion and clarify any concerns you may have. We look forward to your response.

---

### Official Review · Reviewer_fL6t · 2024-11-03

**Soundness:** 3
**Presentation:** 3
**Contribution:** 2
**Rating:** 8
**Confidence:** 4

**Summary:**

This paper introduces a new model, RODIN, for 3D vision language understanding. RODIN leverages the powerful ODIN's backbone and extends it with a proposed 3D mask-language decoder based on Mask2Former or a text decoder to achieve state-of-the-art performance on multiple down-stream 3D tasks, including visual grounding, open-vocabulary object detection, and 3D QA. Unlike existing models, RODIN can process raw posed RGBD frames directly, increasing its scalability.

**Strengths:**

1. This work presents a unique approach by combining an ODIN encoder and a 3D mask-language decoder or a text decoder to solve 3D vision-language understanding tasks directly on posed RGBD frames.

2. RODIN achieves significant improvements over previous methods on multiple downstream benchmarks.

3. The paper is well-structured, with a clear presentation.

**Weaknesses:**

1. This paper's key idea or contribution is simply extending ODIN with existing mask or text decoder designs for 3D visual language understanding. The main components come from experiences of previous works, making it like an engineering combination, although figuring it out to work with SoTA performance is also appreciated. BTW, the benefit of this method, which can process raw RGB-D input to fit the practical setting, also can not be regarded as a contribution because it comes from ODIN. It is also encouraged to supplement some recent works related to this topic to make a more comprehensive discussion:

- EmbodiedScan: A Holistic Multi-Modal 3D Perception Suite Towards Embodied AI
- LLaVA-3D: A Simple yet Effective Pathway to Empowering LMMs with 3D-awareness

2. In the era of LLMs, I am curious about whether such an idea can be applied to MLLMs and how they perform, which I believe can bring more significant progress or attract more attention than solving these tasks in the previous specialist manner.

3. RODIN relies on datasets with segmentation annotations, however most large-scale datasets do not provide segmentation annotations, which may limit the model's scalability. The author discusses this point in Discussion, but it is not addressed.

4. This paper does not discuss the computational efficiency of RODIN, including hardware requirements and inference time. For real-world applications like robotics, efficiency is a critical factor.

**Questions:**

1. The paper of ODIN mentions that "Inaccurate depth or camera poses cause a sharp decrease in performance." How do these affect the performance of the RODIN?

---

> ### Author Response · Authors · 2024-11-21
>
> Thank you for your constructive feedback, insights and support for our work. We address your concerns point-by-point below.
>
> **Q3.1**
> > This paper's key idea or contribution is simply extending ODIN with existing mask or text decoder designs for 3D visual language understanding. The main components come from experiences of previous works, making it like an engineering combination, although figuring it out to work with SoTA performance is also appreciated.
>
> As you mentioned, figuring out these engineering innovations and attaining clear SOTA performance was non-trivial. Additionally, many impactful works are essentially engineering innovations -- MAE is BERT for the visual domain, MaskFormer is DETR with mask decoding heads, ODIN is Mask2Former with additional 3D attention layers, etc. Similarly, RODIN relies on generic architectures like ODIN and Mask2Former, and extends it to 3D vision language tasks. We outperform prior SOTA (with often complex architectures and more engineering) by more than 10% across several benchmarks, which highlights the importance of our technical contributions / engineering.
>
> \
> **Q3.2**
> > The benefit of this method, which can process raw RGB-D input to fit the practical setting, also can not be regarded as a contribution because it comes from ODIN.
>
> ODIN made that contribution **for instance segmentation but not for 3D referential grounding**. Its performance on referential expression is 24.2% worse than RODIN (Table-5, L477-481). We additionally benchmark prior SOTA methods by using sensor point clouds on referential grounding tasks (5-10%) and question answering tasks (negligible difference). RODIN is the first successful instantiation of a model that operates over sensor point clouds and does not use unrealistic assumptions of GT bounding box proposals. We believe that benchmarking in these realistic settings and showing a successful instantiation of a sensor based model like RODIN will make it easier for future works to make progress in settings more aligned to goals of embodied vision.
>
> \
> **Q3.3**
> > It is also encouraged to supplement some recent works related to this topic to make a more comprehensive discussion: EmbodiedScan and Llava-3D.
>
> Thank you for pointing these out -- we have added discussion of these in our revised version.
>
> \
> **Q3.4**
> > In the era of LLMs, I am curious about whether such an idea can be applied to MLLMs and how they perform, which I believe can bring more significant progress or attract more attention than solving these tasks in the previous specialist manner.
>
> This is certainly an interesting direction. As you point out, these ideas can be combined with LLMs and would likely help. For example, LISA [1] uses LLMs as well as SAM decoder heads for grounding. This is an interesting avenue for future research.
>
> [1]: Lai, Xin, et al. "Lisa: Reasoning segmentation via large language model. arXiv (2023)." arXiv preprint arXiv:2308.00692 (2023).
>
> \
> **Q3.5**
> > RODIN relies on datasets with segmentation annotations, however most large-scale datasets do not provide segmentation annotations, which may limit the model's scalability. The author discusses this point in Discussion, but it is not addressed.
>
> You are correct, and as discussed, mask decoding models can have data bottlenecks w.r.t. segmentation annotations. We mentioned Box2Mask in our discussion which achieves strong performance on segmentation tasks by converting box labels to masks via a set of heuristics. They show strong results on ScanNet and ArkitScenes, which suggests that box annotations can work equally well for mask decoding heads, and thus should not be considered as a major limitation while deciding between box vs mask decoders. As our evaluations are based on the ScanNet dataset which was also used by Box2Mask, one would likely obtain similar conclusions as Box2Mask and hence we decide not to spend extra effort on it in this work. As we mention in the discussion, mask decoding has several additional benefits like better performance, more fine-grained output, and unified interface for 2D and 3D tasks.

---

> ### Author Response · Authors · 2024-11-21
>
> **Q3.6**
> > This paper does not discuss the computational efficiency of RODIN, including hardware requirements and inference time. For real-world applications like robotics, efficiency is a critical factor.
>
> Thank you for pointing this out. Per your suggestion, we tested it and found that RODIN has an inference time of 1100 ms and peak memory usage of 15 GB VRAM on an A100 given a scene with 90 images—the average on ScanNet. We note that the memory can be reduced with batching in the visual encoder as the peak memory is consumed by layers that do not have interdependence between images. We further observe that the inference time is roughly proportional to the number of frames and faster inference can be obtained with a variety of common techniques such as motion-based frame selection and optimization techniques for transformer-based models. We have added a description of this performance to the implementation details section.
>
> **Q3.7**
> > The paper of ODIN mentions that "Inaccurate depth or camera poses cause a sharp decrease in performance." How do these affect the performance of the RODIN?
>
> To answer your insightful question, we added an experiment to systematically study the robustness of RODIN and baselines, under noise in both poses and in depth. We evaluate our method and baseline on SR3D and increase the noise until both methods fully fail. We observe that RODIN is significantly more robust to error in both pose and depth compared to prior methods (Figure-6 of supplementary).

---

> > ### Comment · Reviewer_fL6t · 2024-11-22
> > **Final Decision**
> >
> > Thanks for the author's feedback, which addresses most of my concerns. After reading other reviews and the author's rebuttal, I believe the key point is how we appreciate the simple idea (although the implementation that makes it work may be non-trivial) that makes such a state-of-the-art work. From my perspective, I think it is beneficial to the community with the comprehensive experimental results as support, and does provide a solid baseline for this task. Therefore, I would keep my original rating (but maybe a weak accept between 6 and 8).

---

### Official Review · Reviewer_hvvq · 2024-11-03

**Soundness:** 3
**Presentation:** 3
**Contribution:** 2
**Rating:** 6
**Confidence:** 2

**Summary:**

The authors proposed RODIN in this paper, which is a text-RGBD multimodal model for 3D vision language understanding. RODIN extends ODIN [1] to be able to reason with open-vocabulary texts. It uses the pretrained RGBD encoder in ODIN and a pretrained CLIP language encoder to encode RGBD images and texts. It follows Mask2Former’s mechanism to use a set of object queries and a transformer-based query refinement module. In this module, the authors propose to also update the features from RGB-D encoder with cross-attention and show performance gain in 3D referential grounding. The authors evaluate the proposed model in multiple datasets and benchmarks and show strong performance, especially when using RGB-D data directly from the sensors without mesh postprocessing.


[1] Ayush Jain, Pushkal Katara, Nikolaos Gkanatsios, Adam W Harley, Gabriel Sarch, Kriti Aggarwal, Vishrav Chaudhary, and Katerina Fragkiadaki. Odin: A single model for 2d and 3d segmentation. In Proceedings of the IEEE/CVF Conference on Computer Vision and Pattern Recognition, pp. 3564–3574, 2024.

**Strengths:**

- The paper is overall well-written and easy to follow.
- The related work provides a comprehensive overview of 3D visual language understanding.
- The overall design of RODIN is sound and valid.
- The authors demonstrate that the model achieves state-of-the-art performance on several 3D language grounding datasets and benchmarks with a large margin.
- The ablation studies presented in Tables 4 and 5 are sound and well-designed, effectively demonstrating the effectiveness of each design component, particularly the visual token updating mechanism in the query refinement module.

**Weaknesses:**

- The model designs are not new: it mostly extends the prior work ODIN to open vocabulary, and follows Mask2Former's decoding design. The architecture novelty is thus limited - most noticable change is the visual feature updating scheme in the query refinement module.
- The fairness in comparison:
    - are the models shown in Table 1 trained on the same set of data? In L323, the authors mentioned that "This is similar in scale to datasets used by prior SOTA methods like PQ3D Zhu et al. (2024) and 3DVista Zhu et al. (2023b)." Could the authors elaborate more on this?
    - In terms of model size, are the models shown in Table 1 of similar model size?

**Questions:**

- What do each of the losses terms in equation 7 refer to?
- On eq6, what if a none phrase consists of several words?
- typos:
    - L263: "do not update the visual features through during query refinement as we do" is not grammatically sound.
    - L272: naswers -> answers
    - eq 5 & 6: what does $V^T$ mean?
    - eq 6: missing parentheses around $V^T$
    - L286: $f\theta$ -> $f_\theta$

---

> ### Author Response · Authors · 2024-11-21
>
> Thank you for your constructive feedback and helpful suggestions. We address your concerns point-by-point below.
>
> **Q2.1**
> > The model designs are not new: it mostly extends the prior work ODIN to open vocabulary, and follows Mask2Former's decoding design. The architecture novelty is thus limited \- most noticable change is the visual feature updating scheme in the query refinement module.
>
> RODIN’s proposed mask decoder is new and critical to 3D referential grounding performance. Indeed, as you mention, a key change is the visual updating scheme in query refinement. While ODIN's mask decoder does not work for 3D referential grounding (-24.1%, L477-481), this change is crucial to make it work for 3D referential grounding. As we argue in the common response, many impactful works are seemingly simple changes and applying generic architectures to new domains \-- MAE is essentially BERT for the visual domain, MaskFormer is DETR with mask decoding heads, ODIN is Mask2Former with additional 3D attention layers, etc. Similarly, RODIN relies on generic architectures like ODIN and Mask2Former, and extends it to 3D vision language tasks. It outperforms prior SOTA (with often complex architectures and more engineering) by more than 10% across several benchmarks, which highlights the importance of our technical contributions.
>
> In addition, we add a box loss which adds an additional 4.2% accuracy improvement. RODIN is also the first model which directly decodes masks, unlike SOTA 3D referential grounding models which are two-stage. Additionally, we benchmark RODIN and prior SOTA baselines on more realistic settings for robotics/embodied AI and demonstrate even larger performance gains in such real-world settings.
>
> \
> **Q2.2**
> > Fairness in comparison: Are the models shown in Table 1 trained on the same set of data compared to PQ3D and 3D Vista? Could the authors elaborate more on this?
>
> RODIN uses significantly less training data than prior works of PQ3D and 3D Vista. Below are the complete datasets for each:
>
> - 3D VISTA is trained on: SR3D, NR3D, ScanRefer, ScanNet200, ScanQA and SQA3D, 3RScan (1500 scenes), Objaverse (700k objects), text sentences on ScanNet generated using GPT-3 (see Table-3 of 3D Vista \[1\])
> - PQ3D is trained on: SR3D, NR3D, ScanRefer, ScanNet200, ScanQA and SQA3D, Multi3DRefer,  Scan2Cap and a point encoder pre-trained on all data of 3D-Vista.
> - RODIN is trained on: SR3D, NR3D, ScanRefer, ScanNet200, ScanQA and SQA3D, Matterport 3D detection dataset (90 scenes)
>
> \
> **Q2.3**
> > Fairness in comparison**:** In terms of model size, are the models shown in Table 1 of similar model size?
>
> Yes. RODIN has around 130M trainable parameters, while PQ3D has around 247M trainable parameters and 3D-Vista has around 137M trainable parameters. RODIN and PQ3D use frozen CLIP as text encoders. Additionally, PQ3D uses various image/point/voxel backbones that are either fully frozen, or pre-trained and then frozen in a 2nd stage.
>
> We also note that we can achieve optimal performance with far fewer trainable parameters by relying heavily on pre-trained backbones. For example, in Table-6 of supplementary, RODIN with frozen backbone (38M trainable parameters) works similarly to RODIN with fine-tuned backbone (130M trainable parameters). Additionally, we note that PQ3D and 3D Vista are multi-stage models and rely on external models for intermediate outputs like detected objects.
>
> \
> **Q2.4**
> > What do each of the loss terms in equation 7 refer to?
>
> $\\mathcal{L}_{\\text{mask}}$ is the mask loss comprised of Binary Cross Entropy and Dice Losses, $\\mathcal{L} _{\\text{text}}$ is the loss for matching the object queries to the mentioned objects in the language sentence, $\\mathcal{L} _{\\text{gen}}$ is the cross-entropy loss over the auto-regressively generated answer (in case of question-answering datasets), and $\\mathcal{L} _{\\text{box}}$ is the additional bounding box loss (L290-298).
>
> Thank you for pointing out that this was not clear -- we added a clarification text in L303-307.
>
> \
> **Q2.5**
> > On eq6, what if a none phrase consists of several words?
>
> In these cases, the model is trained to output high probability over all words of the relevant noun. The loss is binary cross-entropy over each token, and hence multiple tokens can be the ground-truth target.
>
> \
> **Q2.6**
> > Typos
>
> Thank you\! We fixed them.
>
> \
> **Q2.7**
> > What does $V^T$ mean in eq 5 and 6?
>
> V denotes the visual features, and $V^T$ is the transpose of the V matrix.

---

> ### Author Response · Authors · 2024-11-26
>
> Thank you for your response and engaging in discussion with us. We would like to discuss further on your comment on the limited algorithm contribution of RODIN.  As we mention in our original response to yourself and other reviewers, the changes we did (attention from visual features to queries and box losses) were non-trivial to us and crucial to the performance. 3D referential grounding is a very active field of research with dozens of papers published on these benchmarks every year -- had these been minor or trivial changes, this baseline would already exist by now.
>
> We also want to draw your attention to other contributions of this work besides the technical contributions. As other reviewers also note, RODIN is the first successful instantiation of a 3D language grounding model that operates over sensor point clouds and does not use unrealistic assumptions of GT bounding box proposals. We believe that benchmarking in these realistic settings and showing a successful instantiation of a sensor based model like RODIN will make it easier for future works to make progress in settings more aligned to goals of embodied vision. RODIN is also the first end-to-end trainable 3D referential grounding method with SOTA performance, bypassing prior methods with multi-stage training.
>
> In the broader context of 3D language grounding, RODIN might potentially have important implications. Overall, the current trend in the 3D language grounding community is to:
> a) Encode 3D point cloud directly without using 2D features
> b) Operate over reconstructed and post-processed mesh point clouds
> c) Assume access to GT proposals and / or operate in two-stages i.e. first detect boxes and then select the correct proposal
> or d) Not assume GT boxes and operate in single-stage but decode boxes and obtain low performance on strict thresholds.
>
> RODIN challenges these standard practices and obtains strong performance over all prior methods. We kindly ask you to re-examine your rating / decision in context of both technical as well as these broader contributions of our work. We are very happy to engage in discussion and clarify any concerns you may have. We look forward to your response.

---

### Official Review · Reviewer_3mQZ · 2024-11-04

**Soundness:** 3
**Presentation:** 3
**Contribution:** 2
**Rating:** 6
**Confidence:** 4

**Summary:**

This paper presents a novel 3D vision language model RODIN that directly operates on posted RGBD images rather than sensor point clouds and it argues that the sensor point cloud leads to 5~10% drop in 3D grounding tasks. RODIN extends ODIN model's architecture by adding text encoding/decoding capabilities for language understanding tasks. The whole model is end-to-end trainable. It achieves state-of-the-art performance on multiple 3D vision-language benchmarks including 3D grounding, segmentation, and vision question answering.

**Strengths:**

1. The performance improvement is significant. RODIN outperforms previous state-of-the-art by more than 10% on SR3D, NR3D, and  ScanRefer.
2. The authors show comprehensive ablation studies validating the architectural designs.
3. This work directly work on the sensor data, without requirements for mesh reconstruction.
4. The whole model is end-to-end trainable and can solve multiple 3D vision-language tasks.
5. This architecture can fully leverage the pre-trained 2D vision foundation model features.

**Weaknesses:**

1. The contribution might be limited as the main model architecture is based on ODIN, with the author incorporating text encoding and decoding to add text capabilities.
2. Since it depends on pre-trained 2D models, this approach might limit the model’s ability of generalization. Have authors fine-tuned these pre-trained 2D models, and would fine-tuning enhance the model’s performance?
3. How is camera pose inputted into the model? It’s unclear how the model handles noise in the camera pose. Why is this design performance better than using sensor point clouds, especially given that both designs need to address sensor noise?

**Questions:**

1. Please provide more details about the “posed” RGB-D images. It’s unclear how the camera pose is encoded for each image.
2. How well does RODIN generalize to various camera configurations? If the test set includes different camera setups, will the model still perform effectively?

---

> ### Author Response · Authors · 2024-11-21
>
> Thank you for your constructive feedback and suggestions for experiments. We address your concerns point-by-point below.
>
>
> **Q1.1**
> > The contribution might be limited as the main model architecture is based on ODIN, with the author incorporating text encoding and decoding to add text capabilities.
>
> We use ODIN as our visual encoder, however, that is not our main contribution, neither is adding text encoding and decoding. We add an additional attention from visual features to object queries – this attention layer does not exist in any prior architectures of Mask2Former / ODIN / BUTD-DETR. This change is critical and improves referential grounding performance by 24.2% over a straightforward application of ODIN’s mask decoder (Table-4, L477-481). We additionally propose a box loss which boosts performance by 4.2%. Besides these technical contributions, as you also clearly articulated, RODIN's performance improvement is significant (more than 10%), operates directly over sensor point clouds, is end-to-end trainable for several 3D vision-language tasks (while existing SOTA is two-stage), and fully leverages pre-trained 2D vision foundation features.
>
> \
> **Q1.2**
> > Since it depends on pre-trained 2D models, this approach might limit the model’s ability of generalization. Have authors fine-tuned these pre-trained 2D models, and would fine-tuning enhance the model’s performance?
>
> We finetune all 2D pre-trained weights for the results in the main paper. Additionally, prior works \[1, 2\] show that pre-trained 2D features help generalization in general instead of limiting it. Motivated by your question, we added an experiment in Table-6 where we compare versions of RODIN with and without fine-tuning 2D parameters. We train on SR3D and NR3D, and test on the val sets of SR3D and NR3D (in-domain) and ScanRefer (out-of-domain). We see similar performance both in-domain and out-of-domain, irrespective of freezing or fine-tuning 2D weights. We note however, that since 3D scenes for all these datasets come from ScanNet, it is inconclusive whether fine-tuning 2D features help or hurt generalization for out-of-domain visual 3D scenes (this will be interesting to test, however, that is out of scope for this work).
>
> \[1\]: Yang, Lihe, et al. "Depth Anything V2." *arXiv preprint arXiv:2406.09414* (2024).
> \[2\]: El Banani, Mohamed, et al. "Probing the 3d awareness of visual foundation models." *Proceedings of the IEEE/CVF Conference on Computer Vision and Pattern Recognition*. 2024\.
>
> \
> **Q1.3**
> > How is camera pose inputted into the model? It’s unclear how the model handles noise in the camera pose. Why is this design performance better than using sensor point clouds, especially given that both designs need to address sensor noise?
>
> RODIN does not directly take camera pose as input, but takes a pointmap, which we get from camera intrinsics \+ depth by unprojecting 2D features to 3D. This 3D point information is used in the 3D cross-view attention layers present in the ODIN backbone and for positional embeddings in the mask decoder. Thus camera noise affects the quality of the point cloud. RODIN achieves SOTA results on these noisy sensor point clouds. We believe this is because the 2D backbone operates over clean RGB images directly, and thus adds more robustness to noisy point clouds.
>
> Motivated by your insightful comment, we added an experiment to systematically study the robustness of RODIN and baselines to sensor noise. We evaluate our method and baseline on SR3D and increase the noise until both methods fully fail. We observe that RODIN is significantly more robust to error in both pose and depth compared to prior methods (Figure-6 of the appendix).
>
> \
> **Q1.4**
> > How well does RODIN generalize to various camera configurations? If the test set includes different camera setups, will the model still perform effectively?
>
> RODIN can generalize to a varying number of cameras and camera models. As camera information is only used for feature unprojection and not as a direct input to the model, RODIN is not dependent on any particular camera model or a particular set of intrinsics or extrinsics. The camera pose sequences can be quite different between training and test time. For instance, each scene in ScanNet is captured by a varied number of images with varied camera motion profiles and RODIN works well.
>
> Additionally, at training time, we only train on 15 posed RGB-D images from a given scene, whereas at test time, we feed as input all RGB-D images of a given scene. Our model generalizes well to these different kinds of camera setups, as evidenced by this large train/test camera setup gap and yet the clear SOTA performance.

---

### Author Response · Authors · 2024-11-21
**Common Respone**

We appreciate the reviewers (3mqz, hvvq, fL6t, mEfc) for their valuable feedback. They recognize RODIN for its significant performance gains, surpassing previous state-of-the-art by over 10% on SR3D, NR3D, and ScanRefer (3mqz, hvvq, fL6t, mEfc), establishing it as the first end-to-end method utilizing 2D features that can solve multiple 3D Visual-Language tasks (fL6t, mEfc). The thorough ablation studies confirm the effectiveness of our architectural choices (3mqz, hvvq). Reviewers highlight the model's direct operation on sensor data without mesh reconstruction and its end-to-end training (3mqz). The paper is noted for its clarity and good structure (hvvq, fL6t), along with a comprehensive review of related work (hvvq).

\
**Reviewers 3mqz, hvvq, fL6t, mEfc are concerned that the proposed model has limited contribution over the past work of ODIN.**

**ODIN is a model for instance segmentation** that, similar to our model, can process posed RGB-D sequences and uses pre-trained 2D foundational features. Our proposed model, **RODIN, is a model for 3D referential grounding and question answering. RODIN starts from ODIN’s backbone, and substantially innovates over its decoders.** Specifically, key architectural innovations of RODIN that are necessary for language grounding include: (1) updating visual features via attention to object query and language tokens in decoder (+24.1%) and (2) addition of a box supervision loss (+ 4.2%). **Using the original design of ODIN for referential grounding results in a 24.1% performance drop compared to RODIN (L477-481)**, which justifies our approach.

Apart from the technical contributions, **we further benchmark RODIN and prior SOTA baselines on more realistic settings for robotics/embodied AI** and demonstrate even larger performance gains in such real-world settings. RODIN is also the first end-to-end trainable 3D referential grounding method with SOTA performance, bypassing prior methods with multi-stage training.

Lastly, we have extended RODIN to use DINOv2 ViT backbones for image featurization, without 3D fusion inside the ViT 2D backbone (like the ODIN strategy), but rather, post 3D fusion after the 2D featurization across views. Using ViT 2D backbones improves performance over the initially reported RODIN results on all benchmarks by $4.2\%$, $1.9\%$, and $3.4\%$ @ 0.25 on SR3D, NR3D, and ScanRefer, respectively. Hence, the key message of RODIN is that using 2D features significantly boosts performance for 3D visual language tasks, and it is not restricted to the ODIN backbone.



### **New Experiments since Submission**

Motivated by the reviewers’ comments, we have updated openreview with the revised submission (changes highlighted in magenta). In addition to responses to specific reviewers, below we summarize the major revisions and additional experiments we have added to the paper, guided by reviewers' comments:
- We have included CIDEr/BLEU Metrics on ScanQA in Table-8 of the paper (mEfc), details to architectural diagrams in Figure-5 (mEfc), and clarified details on the loss function and inference performance (hvvq).
- We have measured robustness of RODIN and baselines against varying amounts of sensor noise  in Figure-6 (3mqz, hvvq)
- We reworded “open-vocabulary” object detection to “language-prompted” object detection (mEfc).
-  We tuned hyperparameters and managed to further close the performance gap between RODIN and ODIN in the object detection task (from ~4% mAP lower at the time of submission to only ~1% mAP lower currently), as pointed out by the reviewer mEfc. Currently, RODIN is the best performing model that can both detect objects in 3d and ground open vocabulary language.
- We investigate the effects of fine-tuning 2D backbones on downstream performance as shown in Table-6 (3mQz).

---

### Meta-Review · Area_Chair_rNoq · 2024-12-22

**Metareview:**

This paper presents RODIN, a model that integrates 2D foundational features for 3D vision-language tasks, leveraging posed RGB-D data without mesh reconstruction. It builds on ODIN’s architecture but introduces modifications to enable referential grounding, question answering, and object detection. RODIN achieves SOTA performance on several benchmarks (e.g., SR3D, NR3D, ScanRefer). Reviewers raised several concerns about the difference from ODIN and some questions regarding details. Most concerns are addressed in the rebuttal process, however, the concern that the proposed model has limited contribution over ODIN still remains. Reviewer mEfc finds that "ODIN serves as a new baseline with good performance does not validate the significance or generality of RODIN’s modification. The modification still appears relatively minor, as it builds on ODIN, which already supports multi-modal prompts. If it is a generic 2D operation that enhances 3D features, additional experiments on 3D backbones, such as PointTransformer or Minkowski (as mentioned by the authors), would provide stronger evidence of its significance." AC carefully reads ODIN and RODIN, and agrees that the methodological difference from ODIN is not significant, while AC does acknowledge the engineering designs and their corresponding improvements. That said, AC believes the paper still does not meet the standard of ICLR due to the missing of adequately intellectual novelty. Therefore, AC recommends a weak rejection.

**Additional Comments On Reviewer Discussion:**

As mentioned above, most concerns regarding technical details have been addressed in the rebuttal period. However, a critical concern, i.e. methodological difference from ODIN, still remains. Reviewer mEfc and authors themselves suggested addition experiments on 3D backbones, such as PointTransformer or Minkowski, were not provided to justify authors' claims.

---

### Decision · Program_Chairs · 2025-01-22

Reject